# Simultaneous Improvement in Mechanical Properties and Fatigue Crack Propagation Resistance of Low Carbon Offshore Structural Steel EH36 by Cu–Cr Microalloying

**Xingdong Peng** [1,2], **Peng Zhang** [2,*], **Ke Hu** [1,*] , **Ling Yan** [2] **and Guanglong Li** [2]

1   School of Materials and Metallurgy, University of Science and Technology Liaoning, Anshan 114051, China; pxd197395@163.com
2   State Key Laboratory of Metal Material for Marine Equipment and Application, Anshan 114001, China; yanling_1101@126.com (L.Y.); liguanglong@ansteel.com.cn (G.L.)
*   Correspondence: zp15842019779@163.com (P.Z.); hukee12345@126.com (K.H.)

**Abstract:** Improving the mechanical performance of low-carbon offshore steel is of great significance in shipbuilding applications. In this paper, a new Cu-Cr microalloyed offshore structural steel (FH36) was developed based on EH36. The microstructure, mechanical properties, and fatigue crack propagation properties of rolled plates of FH36, EH36, and normalizing rolled EH36 plates (EH36N) manufactured by a thermo-mechanical control process (TMCP) were analyzed and compared (to simplify, the two rolled specimens are signified by FH36T and EH36T, respectively). FH36T showed an obvious advantage in elongation with the value of 29%, 52.2% higher than the EH36T plates. The normalizing process led to a relatively lower yield stress (338 MPa), but substantially increased the elongation (33%) and lessened the yield ratio from 0.77 to 0.67. Electron back-scattered diffraction (EBSD) analysis showed that SFs of the deformation texture of FH36T and EH36N along the transverse direction (TD) and normal direction (ND) were much higher than those of the EH36T plate, which enhanced the lateral movement ability in the width and thickness direction, enhancing the ductility. Moreover, FH36 plates showed a better fatigue crack propagation resistance than rolled EH36 plates. The formation of the jagged shape grain boundaries is believed to induce a decrease of effective stress intensity factor during the fatigue crack propagation process.

**Keywords:** offshore structural steel; TMCP; EH36; mechanical properties

## 1. Introduction

In recent years, many strategies have been under development for large-scale, lightweight, environmentally friendly ocean vessels with deep ocean accessibility. Marine ships, especially large ocean-going ships, are often confronted with strong winds and waves during their navigation. This often exposes the hull to complex stress conditions due to the huge impact forces and periodic alternating loads from different directions. To ensure the safety and work-reliability of ocean vessels, major steel manufacturers are only focusing on improving the coupling performance of strength, ductility, and fatigue resistance of offshore structural steels [1,2].

As a typical low carbon bearing high strength offshore steel, EH36 possesses various advantages which make it widely used in the marine industry, such as high strength, comprehensive performance, and the ability to reduce the dead weight of the hull. EH36 is a kind of microalloyed low carbon steel with Mn as the main alloying element.

The fine-tuning of the composition and the corresponding production technology have a dramatic effect on the above-mentioned mechanical properties. In terms of alloying strategy, carbon content is undoubtedly the most important consideration due to its significant strengthening and hardening effects. However, recently the focus of offshore structural steel has shifted to cost-effective alternatives by reducing carbon additions,

because low carbon improves the formability and toughness of the steel plate, which facilitates the execution cold forming operations such as stamping and bending [1]. To remedy the decrease of mechanical properties caused by the low carbon content, some attempts of microalloying have been made. Zou et al. [3] found that the addition of a small amount of Mg can obviously refine the grain size of rolled EH36 plates and promote the nucleation of acicular ferrite. They then reported that the trace addition of Zr can promote the fine MnS precipitates in the continuous casting slab of EH36, and significantly enhance the strength, with the yield strength increasing from 390 MPa to 489 MPa, and the tensile strength increasing from 445 MPa to 563 MPa [4]. Ma et al. [5] reported that the addition of Nb can greatly improve the ultimate tensile strength of EH36, which can reach 585 MPa, but the effect on yield strength is not obvious. The improvement in the fatigue crack growth performance of low carbon steel mainly depends on the modification of the microstructure of matrix; e.g., Y. Zhao et al. [6] reported that the lath martensite in low carbon steel formed after TMCP treatment has a uniform local stress concentration during the crack propagation process, which increases the crack growth resistance. Shohei Ueki et al. [7] further revealed that the microstructural inhomogeneity of the low carbon steel enhanced the strain localization in coarse laths, which probably led to the premature fatigue crack growth. There are few studies on improving the fatigue crack growth performance of EH36 through microalloying.

Among the typical alloying elements, Cu and Cr are alloying elements to improve the mechanical performance of rolled steel. It is widely known that C is added in steel to improve hardenability; Cr is an ideal element to replace C and improve the corrosion resistance [8,9]. Copper, on the one hand, is usually perceived as a strong solution-strengthening element, while on the other hand, Sung-Joon K [10] reported that its presence can expand the austenite region on the phase diagram, thus increasing the austenite volume fraction. This indicates a drop in the A3 line by some degrees, which is equivalent to reducing the rolled temperature in a disguised way, hence assisting in refining the final microstructure. Also, Cu-rich precipitates are non-carbide particles, which can provide a substantial increase in strength [11,12]. Therefore, it is a meaningful attempt to further reduce the C content without sacrificing mechanical properties through the synthetic alloying of Cr and Cu.

The manufacturing process also plays an important role in realizing the impact of alloy design on steel, such as heat treatment [13,14] and chemical treatment processes [6,15–17]. Products of low carbon high strength offshore steel mainly adopt a thermo-mechanical control process (TMCP), off-line quenching, and tempering processes [6,18]. TMCP improves the mechanical properties of steels by controlling emulsification and refining austenite grains, introducing processing strain and refining the transformation structure. Furthermore, TMCP allows the amount of alloying to be reduced, which not only improves the strength and toughness of the steel but is also favorable for the welding performance. Owing to the refined grains, TMCP processed steel obtains a kind of strength and low-temperature mechanical properties that cannot be obtained by other heat treatment. However, for a thicker plate, the TMCP will lead to an inhomogeneity in the microstructure from the surface region to the interior of the rolled plate due to the difference in cooling rates. Therefore, the subsequent normalizing process is necessary to ensure the quality of the final product.

Based on the above, this paper aims at improving mechanical properties via adding elements (Cu and Cr) to further suppress the carbon content to lower than 0.06 wt.% to enhance the form ability and weldability. The effect of the corresponding microstructure on tensile performance and fatigue crack propagation properties of the newly developed offshore steel are analyzed in comparison with EH36 manufactured by the thermo-mechanical control process (TMCP) and normalizing. The results can provide theoretical and data support for future alloy design and processing technique development.

## 2. Materials and Methods

### 2.1. Process and Materials

Ingots of EH36 and FH36 were prepared by continuous casting with cross sectional area of 200 mm × 200 mm. Actual chemical compositions are listed in Table 1. The procedure of processing the billets was as follows.

**Table 1.** Chemical composition of the experimental steels (wt.%).

| Elements | C | Si | Mn | Cu | Cr | Ni | V | Ti | Al |
|---|---|---|---|---|---|---|---|---|---|
| EH36 | 0.18 | 0.15 | 1.45 | - | - | 0.40 | 0.04 | 0.015 | 0.040 |
| FH36 | 0.06 | 0.15 | 1.35 | 0.12 | 0.18 | 0.38 | 0.045 | 0.017 | 0.045 |

The cast billets of EH36T and FH36T were first homogenized at 1200 °C for 2 h, followed by rough hot rolling to achieve a thickness of ~80 mm through five passes with a corresponding reduction of 60%. Subsequently, the rolled plates were air cooled to 830 °C for the final rolling process to achieve a thickness of 30 mm through 5 passes with a corresponding reduction of 62.5%. This was followed by water cooling to 480 °C with a measured cooling rate of 15 °C/s, then air cooling to room temperature (Figure 1). The plate of EH36N was only subjected to a rough rolling process similar to that of EH36T and FH36T, followed by reheating to 860 °C with a retention time of 3 h, then air cooling to room temperature.

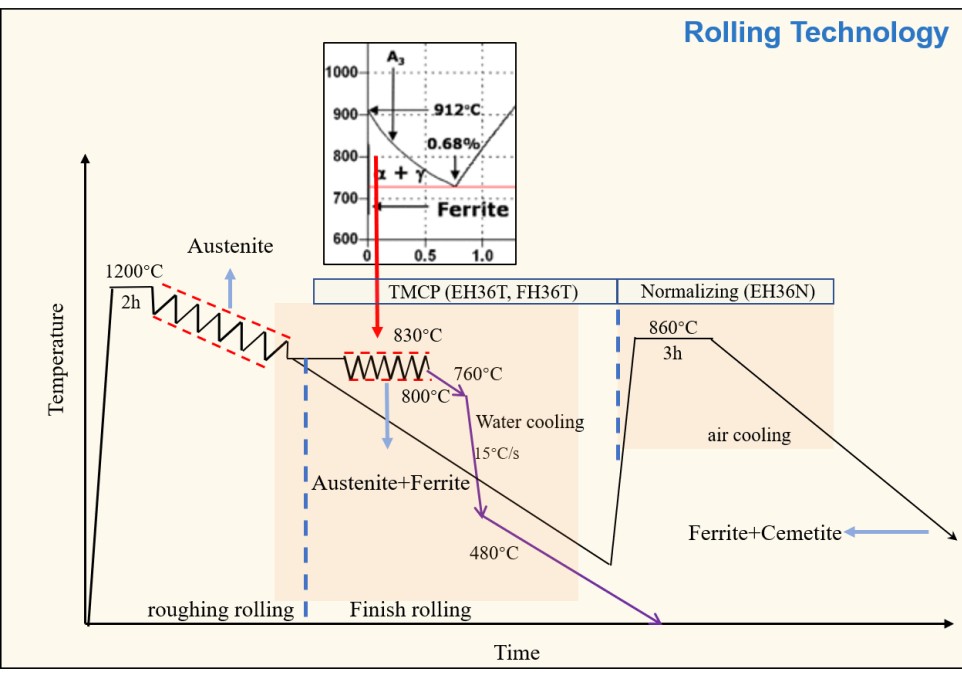

**Figure 1.** Manufacture process of offshore steels.

### 2.2. Mechanical Properties Test

Tensile tests for the base material were performed at room temperature using 12 mm wide and 3 mm thick specimens with a gauge length of 80 mm parallel to the rolled direction in the middle of rolled plate (minimum 3 specimens) as shown in Figure 2. Yield strength (YS), ultimate tensile strength (UTS), and elongation (E) to fracture were determined from the tensile testing results. To ensure the reliability of the experimental data, each tensile test was repeated 3 times at least.

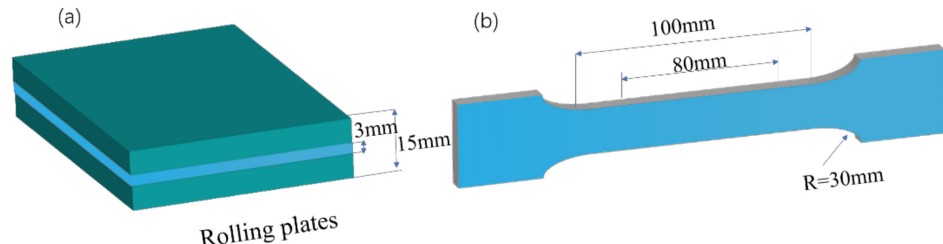

**Figure 2.** Specification of tensile sample: (**a**) rolling plate, (**b**) tensile specimen.

### 2.3. Fatigue Crack Propagation Test

To obtain the Paris formula of the fatigue crack growth rate of offshore steels at room temperature, a fatigue crack growth rate test with increasing stress intensity was carried out based on GB/T 21143-2014 at room temperature. The size of a relevant sample is shown in Figure 3, with the crack propagation direction of the sample along the rolled direction. The room temperature fatigue crack growth rate test was carried out using the INSTRON 8801 hydraulic fatigue testing system (-INSTRON Inc., Boston, MA, USA); the maximum load of the equipment is $\pm 100$ kN. During the experiment, a crack opening displacement (COD) gauge was installed on the sample, and the crack length was measured by the flexibility method. The gauge length of the COD was 5 mm and the maximum opening displacement was 2 mm. Fatigue cracks were firstly prefabricated in room temperature. The loading method was axial loading, with a stress ratio of 0.1, maximum force 16 kN, and test frequency of 20 Hz. The smaller cracks in the plastic zone were obtained by step-by-step load reduction. The constant load control was adopted, and the load reduction amplitude of the adjacent levels did not exceed 20%. Crack propagation length was about 2 mm. Finally, load was kept constant until the sample broke to obtain the $\mathrm{d}a/\mathrm{d}N$ and $\triangle K$. The data points that did not satisfy W-a > 4 $(Kfmax/Rp0.2)^2$ were removed to obtain the double logarithmic curve of $\mathrm{d}a/\mathrm{d}N \sim \triangle K$, and the Paris formula [19] was obtained through double logarithmic linear fitting by Origin 2016 software (OriginLab Inc., Northampton, MA, USA).

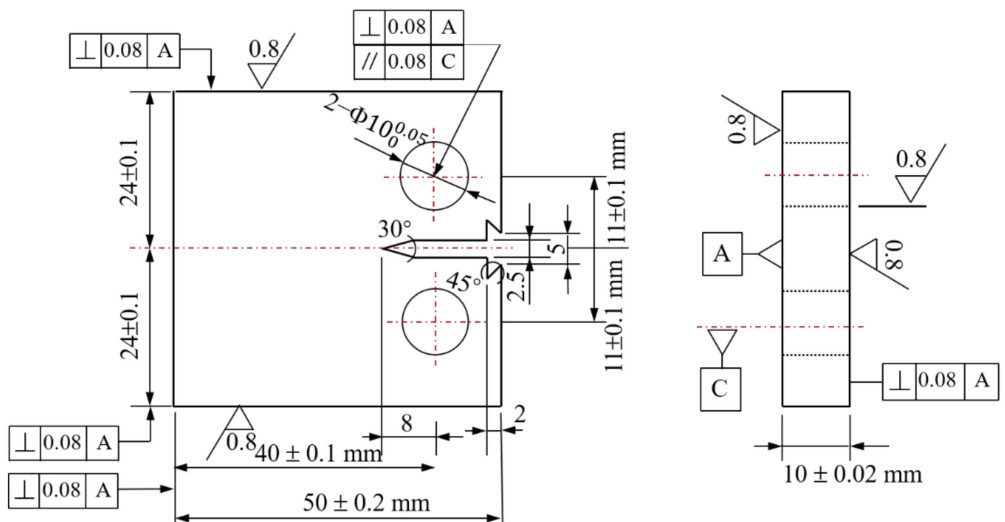

**Figure 3.** Specification of fatigue crack propagation test.

### 2.4. Characterization Methods

The microstructural features of the specimens of EH36T, FH36, and EH36N, such as the morphology and orientation of grains and texture intensity, were characterized through an optical microscope (OM), a scanning electron microscope (SEM), and electron back-scattered diffraction (EBSD). The metallographic specimens with the dimensions of 10 mm × 10 mm × 8 mm were cut perpendicular to the rolled direction (RD). The metallographic

specimen was prepared by progressively increasing abrasive paper with a grit size range of 400–2000# followed by polishing and eroding by 4% nitric acid alcohol. The microstructure was observed by a DM2500M optical metallographic microscope (Leica Microsystems LTD., Weztlar, Germany) and a FEI Quanta 250F field emission environmental scanning electron microscope (FEI Inc., Hillsboro, OR, USA) at room temperature. For the elimination of the specimen surface stress prior to EBSD examination, mechanically ground samples were subjected to electro-polishing with 5% perchloric acid alcoholic solution at ambient temperature with a voltage of 35 V for 10 s. SEM and EBSD method was carried out by a FEI Quanta Tescan MIRA 3 LMH scanning electron microscope (FEI Inc., Hillsboro, OR, USA) equipped with an Oxford EBSD probe (Oxford Instruments Inc., London, UK) at room temperature. The EBSD data was analyzed by TSL OIM 7.3.1 software (EDAX Inc., Mahwah, NJ, USA).

## 3. Results and Discussion

### 3.1. Microstructure Analysis by OM and SEM

Figure 4 shows the microstructures of three samples, which were all composed of ferrite ($\alpha$ phase) and cementite ($Fe_3C$). It can be seen that the EH36T plate had unevenly distributed $Fe_3C$ phases and the phases were mainly distributed within the ferrite grains, presenting a mesh structure as shown in Figure 4a and the corresponding SEM micrograph in Figure 4d. In contrast, the $Fe_3C$ particles in FH36T presented an inconspicuous strip-like shape along the rolling direction at the macro level as shown in Figure 4b. These particles are mainly located at the boundary of ferrite grains. Unlike EH36T, the morphologies of the phases showed a discrete granular shape (Figure 4e). EDS results of position C and D illustrate that the added Cu and Cr were more likely to be dissolved into the $\alpha$-matrix as shown in Table 2. In the case of the EH36N sample, the distribution of secondary phases demonstrated an obvious strip-like shape. At the micro level, the phases were located both within the grains and at the grain boundaries as shown in Figure 4e,f. In addition, the micro-alloying elements were not entirely distributed in the secondary phases. EDS examination reveals that quite a limited percentage of the elements were dissolved into the $\alpha$-matrix.

**Table 2.** EDS results in Figure 4.

| Position | Carbon | Silicon | Aluminum | Nickel | Niobium | Cuprum | Chromium |
|----------|--------|---------|----------|--------|---------|--------|----------|
| A | 6.5 | 0.35 | 0.21 | 0.51 | 0.10 | - | - |
| B | 3.06 | 0.43 | 0.30 | 0.63 | 021 | - | - |
| C | 6.92 | 0.24 | 0.31 | 0.61 | 0.29 | 0.19 | 0.15 |
| D | 3.92 | 0.30 | 0.22 | 0.70 | 0.35 | 0.27 | 0.32 |
| E | 9.57 | 0.28 | 0.31 | 0.57 | 0.15 | - | - |
| F | 2.29 | 0.28 | 0.13 | 0.64 | 0.15 | - | - |

### 3.2. EBSD Analysis

3.2.1. Microstructure Analysis

To closely examine the microstructural features of the material, EBSD analysis was implemented. Figure 5 presents the orientation maps and corresponding distribution of grain boundaries, which shows the morphology of $\alpha$ grains more clearly than an OM image. From the EBSD data, the average grain size of the EH36T, FH36T, and EH36N plates were 7.3 μm, 6.2 μm, and 5.2 μm, respectively. Residual austenite was not detected in any sample. It can be found that the EH36T plates were obviously fully recrystallized. In contrast, the other samples were not completely recrystallized. The dynamic recrystallization occurred both at the grain boundaries and within grains for the FH36T and EH36N plates. In EH36N plates, many newly developed tiny grains can be observed forming at the boundaries of the coarse parent grain, especially at triple grain boundary junctions as shown in the partial enlarged area in Figure 5f.

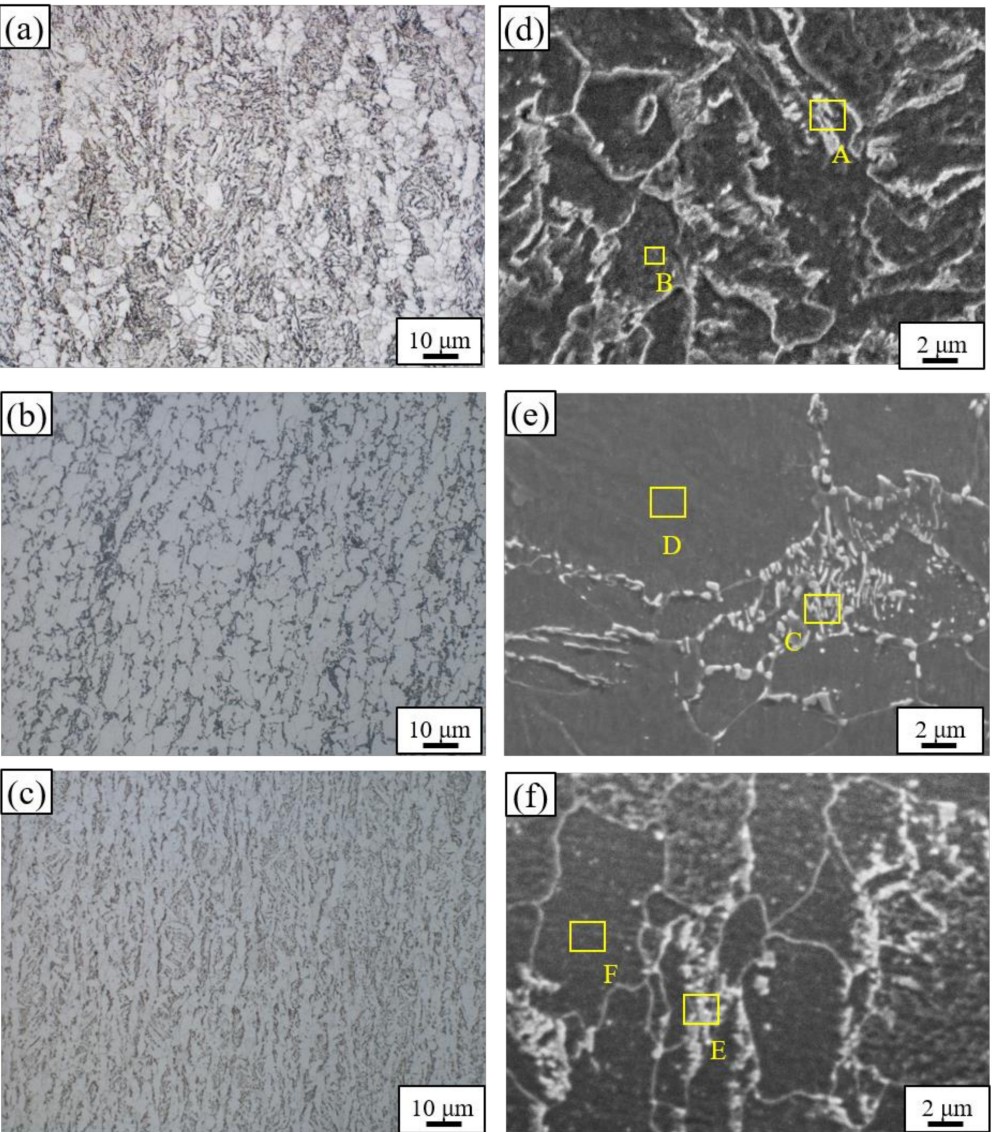

**Figure 4.** Microstructures of offshore structural steels: (**a**,**d**) EH36, (**b**,**e**) FH36T, and (**c**,**f**) FH36N.

As the EH36T and FH36T rolled plates experienced the same deformation process, the reasons for the formation of finial grain structure can be explained through different recrystallization mechanisms. For FH36T, adding Cu can expand the austenite zone [10], thus declining the real A1 line on the iron-carbon phase diagram, which makes the α phases finer. In addition, Cu will generate fine precipitates [20], with diameter less than 1 μm, based on the Zenner effect [21,22]; these dispersed precipitates will inhibit the migration of grain boundaries, inhibiting recrystallization. Therefore, the FH36T plate presented an incompletely recrystallized but mostly refined grain structure.

When it comes to EH36N, due to an absence of TMCP, the degree of grain fragmentation was lower than that of EH36 and FH36. Furthermore, the rolled steels lack "strain" (stored-energy) accumulation; thus, the grain sizes were larger than the other two cases and the degree of recrystallization was the lowest.

### 3.2.2. Textural Analysis

Apart from the grain size, distribution, and morphology of the secondary phases, the textures of metallic materials also play an important role in the mechanical properties. Figure 6 shows the pole figures and corresponding ODF sections of $\varphi_2 = 45°$ ($\varphi_2$, $\Phi = 0–90°$). It can be observed that the pole figure intensity of EH36T was clearly weaker than that of

EH36N and FH36T. To show the distribution of the orientation of grains in the three samples more visually and to calculate the detailed texture, the Euler angles of the three samples were compiled in Figure 7. It is shown that the texture of the EH36T plate presented a typical texture component of ND fiber (orientations that have a <110> axis parallel to the rolled direction), {112}<110>, and {554}<225> components. The detailed textures of the FH36T and EH36N samples were calculated by the following formula:

$$\begin{bmatrix} u & h \\ v & k \\ w & l \end{bmatrix} \begin{bmatrix} cos\varphi_1 cos\varphi_2 - sin\varphi_1 sin\varphi_2 cos\Phi & sin\varphi_2 sin\Phi \\ -cos\varphi_1 sin\varphi_2 - sin\varphi_1 cos\varphi_2 sin\Phi & cos\varphi_2 sin\Phi \\ sin\varphi_1 sin\Phi & cos\Phi \end{bmatrix} \tag{1}$$

where *u*, *v*, and *w* are miller indices of the crystal plane and *h*, *k*, and *l* are the orientation indices.

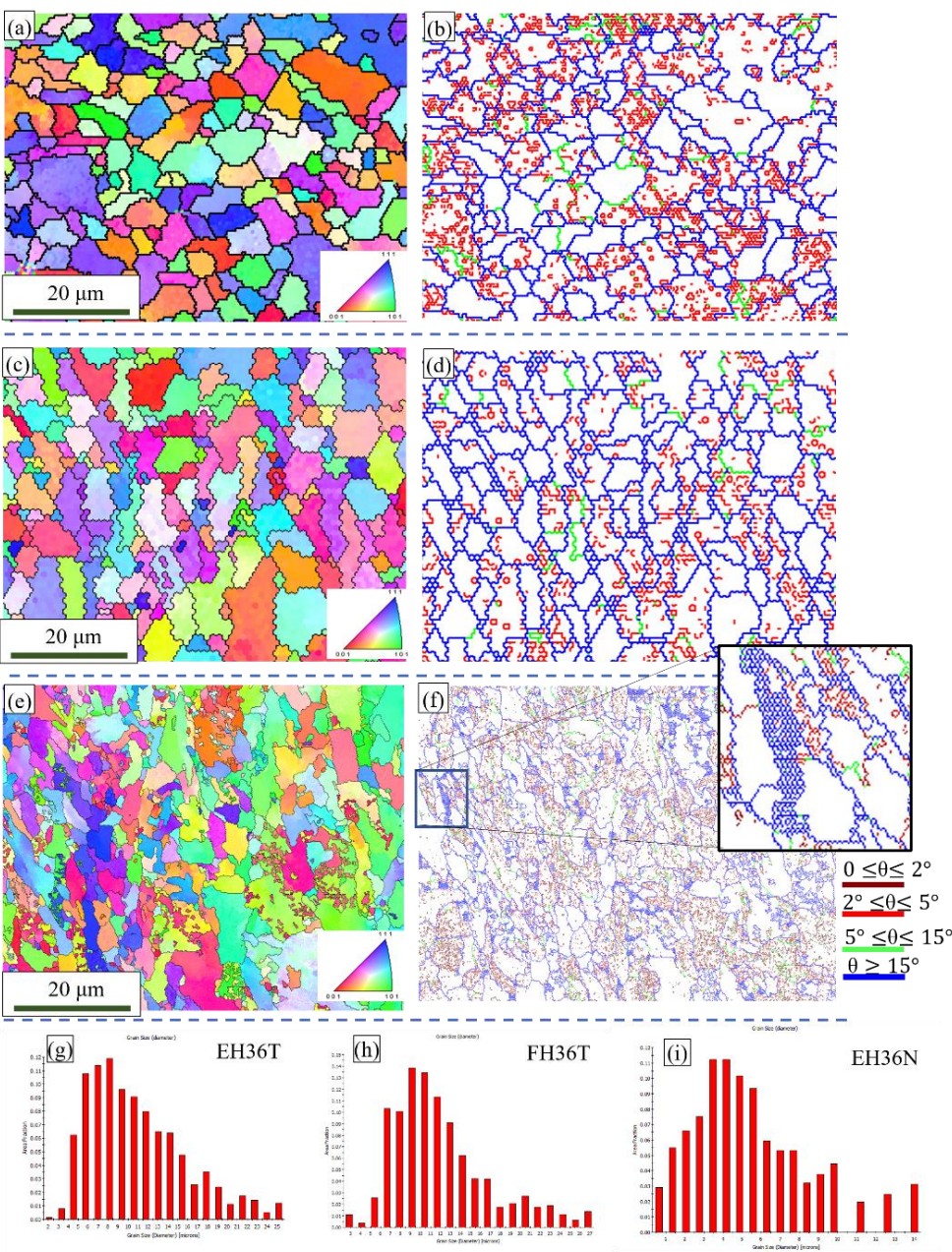

**Figure 5.** Orientation maps and distribution of grain boundaries: (**a**,**d**) EH36, (**b**,**e**) FH36T, (**c**,**f**) FH36N, and (**g**–**i**) histogram rams of frequency of grain boundaries of the offshore steels.

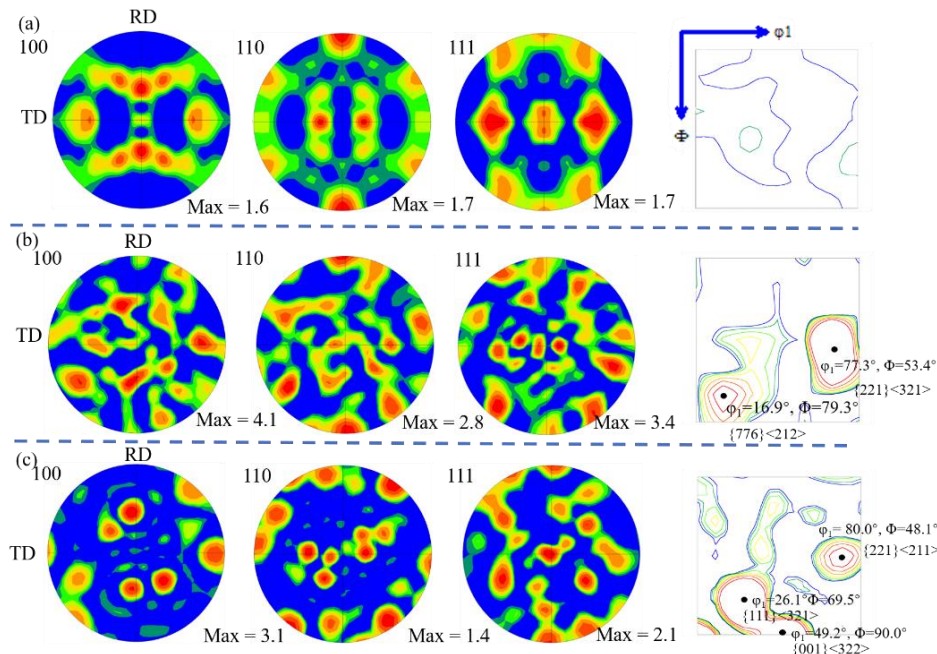

**Figure 6.** Deformation textures of the studied samples represented by pole figures and ODF section: (**a**) EH36T, (**b**) FH36T, and (**c**) EH36N.

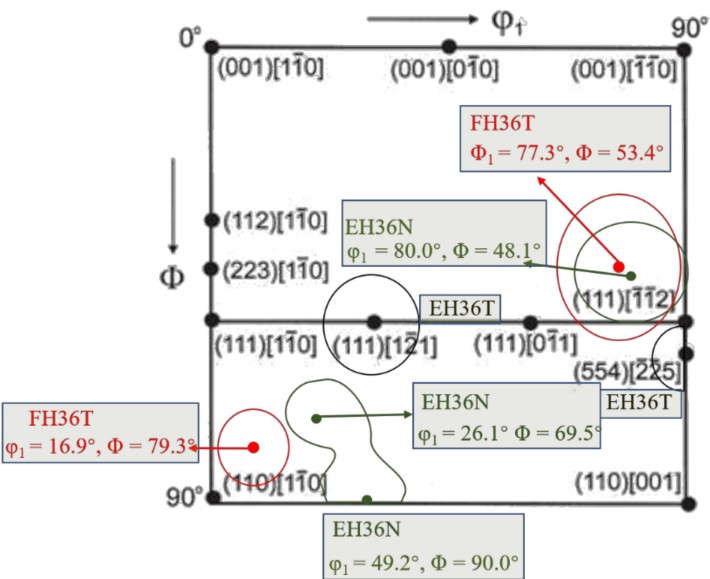

**Figure 7.** ODF sections of $\varphi_2 = 45°$ ($\varphi_1$, $\Phi$ = 0–90°).

Based on the Formula (1), the grain orientations of the FH36T plate mainly comprised two types of deformation texture components, which lay around {221}<321> and {776}<212>. The intensity of the texture was obviously stronger than that of the EH36T specimen while the texture morphology of the EH36N plate was much more similar to that of the FH36T plate, with distribution around {111}<321>, {001}<322>, and {221}<211>, respectively.

### 3.3. Mechanical Performance

#### 3.3.1. Mechanical Properties

Typical stress-strain curves and tensile properties are shown in Figure 8. Related mechanical properties are summarized in Table 3. The FH36T plate demonstrated a better

strength and elongation; however, its yield ratio was the highest among the three samples. EH36N (Rough rolled + normalizing treated EH36 plate) presented an optimum ductility with elongation reaching 33% with an insignificant drop in UTS. Contrary to UTS, yield strength was relatively lower, which led to the lowest yield ratio (YR) of merely 0.67. It is worth noting that FH36T and EH36N presented an obvious yield plateau, which could be accounted for by the pining effect of interstitial atoms, e.g., C, on dislocations [23].

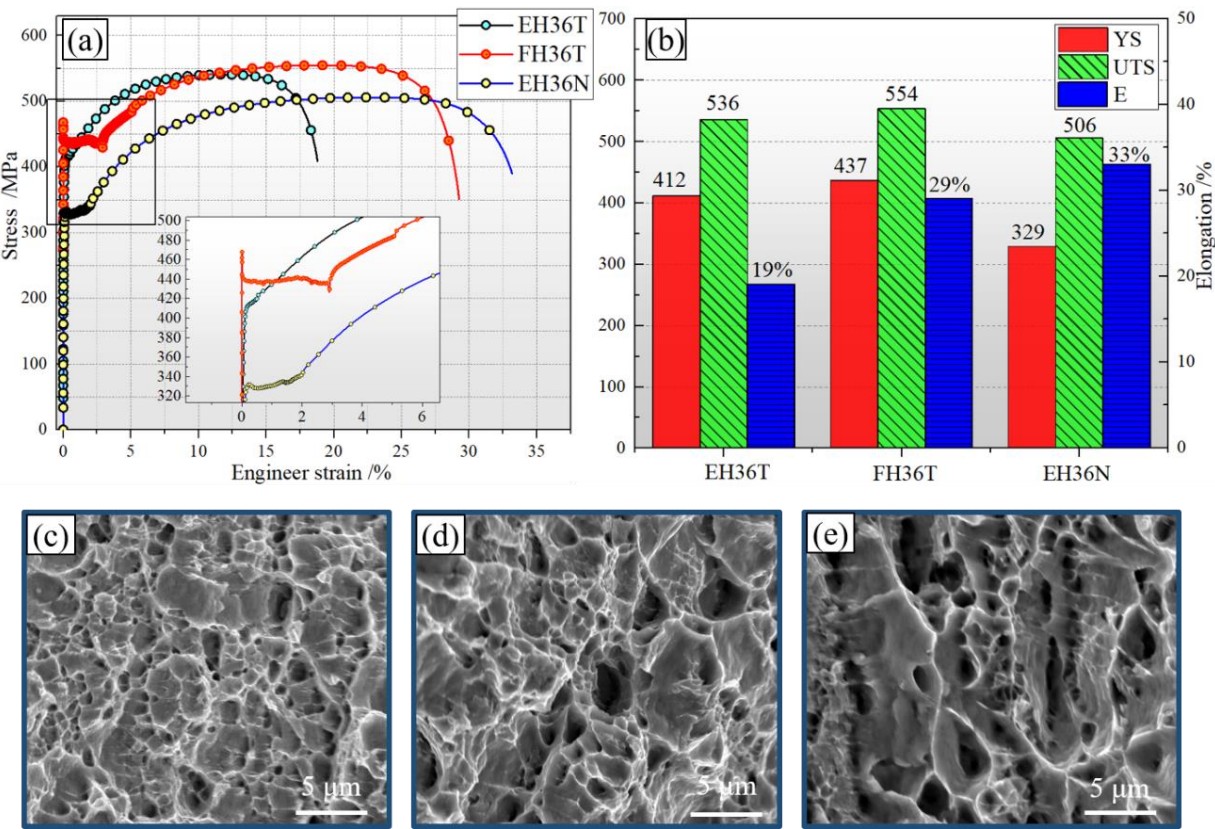

**Figure 8.** Mechanical properties and corresponding fractographs of rolled plate samples: (**a**) stress–strain curves; (**b**) mechanical properties; and (**c–e**) SEM fractographs of the plates of EH36T, FH36T, and EH36N.

**Table 3.** Mechanical properties.

| Sample | YS/MPa | UTS/MPa | E/% | YR |
|--------|--------|---------|-----|-----|
| EH36T | 412 | 536 | 19 | 0.77 |
| FH36T | 438 | 553 | 29 | 0.79 |
| EH36N | 338 | 506 | 33 | 0.67 |

Luo et al. studied the effect of different normalizing treatments on the mechanical properties of TMCP steel; the related mechanical properties are shown in Table 4. They found that a decrease in the normalizing temperature led to a decrease in yield strength [24]. Then, they developed different heat treatment methods to improve the mechanical properties, as shown in Table 4 [25]. Zou et al. [3,4] tried to improve the yield and ultimate tensile strength with Mg and Zr while the elongations were not reported, as shown in Table 4. In comparison with the experiment data of Zou et al. and Luo et al., the EH36T specimen in this study presented a relatively high yield strength, and the FH36T specimen also showed a better balance of strength and elongation. As far as the heat treatment process is concerned, heat preservation time in this study was longer (3 h), which is more suitable for thicker plates. It can be seen that a longer normalizing holding time will not

further improve ductility compared to a shorter holding time (30 min; 1 h), while it will greatly reduce the yield stress without sacrificing more ultimate tensile strength, obtaining a lower YR.

**Table 4.** Mechanical properties of EH36 of relevant studies [3,4,24–26].

| Technological Conditions | Material | YS/MPa | UTS/MPa | Elongation/% | YR | Reference |
|---|---|---|---|---|---|---|
| TMCP | EH36 | 385.2 | 541.1 | 22.7 | 0.71 | [24] |
| 870 °C × 30 min | EH36 | 353.8 | 497.2 | 34.6 | 0.71 | [24] |
| 910 °C × 30 min | EH36 | 372.1 | 494.4 | 35.5 | 0.75 | [24] |
| 950 °C × 30 min | EH36 | 380.6 | 496.3 | 35.2 | 0.77 | [24] |
| 870 °C/30 min + 500 °C/30 min | EH36 | 352.6 | 467.6 | 25.3 | 0.75 | [25] |
| 870 °C/30 min + 550 °C/30 min | EH36 | 371.1 | 493.6 | 27.6 | 0.75 | [25] |
| 950 °C/30 min + 500 °C/30 min | EH36 | 382.6 | 517.2 | 30.6 | 0.74 | [25] |
| 950 °C/30 min + 550 °C/30 min | EH36 | 384.6 | 516.3 | 32.2 | 0.75 | [25] |
| TMCP | EH36 | 420 | 550 | 23 | 0.76 | [26] |
| TMCP (1200 °C × 2 h + 1050–840 °C rolling + 840–440 °C water cooling) | EH36 microalloyed by Mg | 415 | 531 | - | 0.78 | [3] |
| TMCP (1200 °C × 2 h + 1050–840 °C rolling + 840–440 °C water cooling) | EH36 microalloyed by Zr | 445 | 563 | - | 0.79 | [4] |

The fractographs of rolled plate samples are shown in Figure 8c–e It can be determined from the figures that all the cases demonstrated a dimple fracture which is a characteristic ductile fracture. The dimples of the EH36T samples were relatively dense and shallow, in contrast, the ones of EH36N were deeper and distributed unevenly. In most of the dimples, no obvious second phase particles were observed, thus corroborating the fact that the ductility of the samples mainly relied on the microstructure of α-matrix.

### 3.3.2. Strengthening Mechanism

To explain the influence of the microstructure on the strength, structure-property models were proposed [27]. The total yield stress of alloys mainly includes the strengthening effect caused by grain boundaries, dislocations, precipitates, and solid solution strengthening. The contribution of these different strengthening mechanisms to final yield strength can be expressed as follows:

$$\sigma_y = \Delta\sigma_0 + \Delta\sigma_S + \Delta\sigma_{gs} + \Delta\sigma_{Dis} + \Delta\sigma_{ppt} \tag{2}$$

where $\sigma_y$ is yield strength in MPa; $\Delta\sigma_0$ is the friction stress of pure iron (~54 MPa) [28]; $\Delta\sigma_S$, $\Delta\sigma_{gs}$, $\Delta\sigma_{Dis}$, and $\Delta\sigma_{ppt}$ are the contributions of the solid solute, grain boundary, dislocation, and precipitation strengthening, respectively.

The influence of the strengthening effect of grain size on the yielding strength could be estimated by the standard Hall–Patch equation as follows:

$$\Delta\sigma_{gs} = \mathrm{K}d^{-\frac{1}{2}} \tag{3}$$

where K is constant at 17.9 MPa mm$^{1/2}$ [29] and $d$ is the average grain size. This formula was deduced on the basis of the dislocation blockage model. It is believed that the grain boundary is an obstacle to the movement of dislocations, and blockage occurs at the grain boundary during the process of dislocation movement. Li James [30] modified the Hall–Patch formula:

$$\Delta\sigma_{gs} = \mathrm{K}d^{-\frac{1}{2}} + \mathrm{k}G\mathbf{b}(8s/\pi)^{1/2} \tag{4}$$

where $s$ is the density of "ridges" (the number of "ridges" distributed on grain boundaries per unit length or the length of "ridges" per unit area of grain boundaries); k is the undetermined test constant related to dislocation distribution, and taken as approximately 0.4; G is the shear modulus taken as 80.3 GPa; and b is the Burgers vector, having a value of 0.248 nm.

It can be seen that the "ridges" are directly related to the strength of the material. This is because of the fact that a jagged grain boundary can hinder the slip to the neighboring grains and hamper the onset of microcracks formed on the grain boundaries which could subsequently evolve into macroscopic cracks. Thus, it can play a good role in improving strength and plasticity, while also inhibiting temper brittleness and creep rupture. The $\Delta\sigma_{gs}$ for EH36T, EH36N, and FH36T was found to be 66 MPa, 92 MPa, and 79 MPa, respectively.

The contribution of the solid solution strengthening effect can be expressed as follows [31]:

$$\Delta\sigma_S = 360.36[C] + 83.16[Si] + 32.34[Mn] + 40[Cu] - 30[Cr] \tag{5}$$

where [X] is the weight percent of alloying element *X*. It can be calculated from Equation (5) that the $\Delta\sigma_S$ for EH36T, EH36N, and FH36T was 137.4 MPa, 89.2 MPa, and 137.4 MPa, respectively.

The increased yield stress resulting from the increased dislocation density can be estimated by the relationship given as follows [28]:

$$\Delta\sigma_{Dis} = \alpha MG\mathbf{b}\sqrt{\rho} \tag{6}$$

where $\alpha$ is a constant of ~0.435 [28]; $\rho$ is the dislocation density; and M is the Taylor Factor of 2.75 [28]. The dislocation density of ferrite formed by the isothermal transformation was assumed to be $5 \times 10^{13}$ m$^{-2}$ [29]; thus, the $\Delta\sigma_{Dis}$ for every case was 62 MPa. Based on Equation (2), the $\Delta\sigma_{ppt}$ of the EH36T, FH36T, and EH36N can be calculated as 93 MPa, 141 MPa, and 6 MPa, respectively.

In the present paper, no aging treatment was carried out. Except for carbon, the content of most adding elements in this study was less than their solid-solution limits; thus, the water-cooling process will develop a local supersaturated solid-solution zone. Chen reported that even the low carbon steel will experience natural aging effectively. The precipitates including stable $M_{23}C_6$, unstable $M_3C$, Cu-rich phases [20], and the preexisting trace of Ni will promote the precipitation process of Cu-rich phases [31]; thus, the $\Delta\sigma_{ppt}$ of FH36T was the highest. In case of EH36N, the slow cooling process made the local supersaturated solid-solution zone difficult to form. Resultantly, the $\Delta\sigma_{ppt}$ was very limited, which is also consistent with the calculated result as shown in Figure 9.

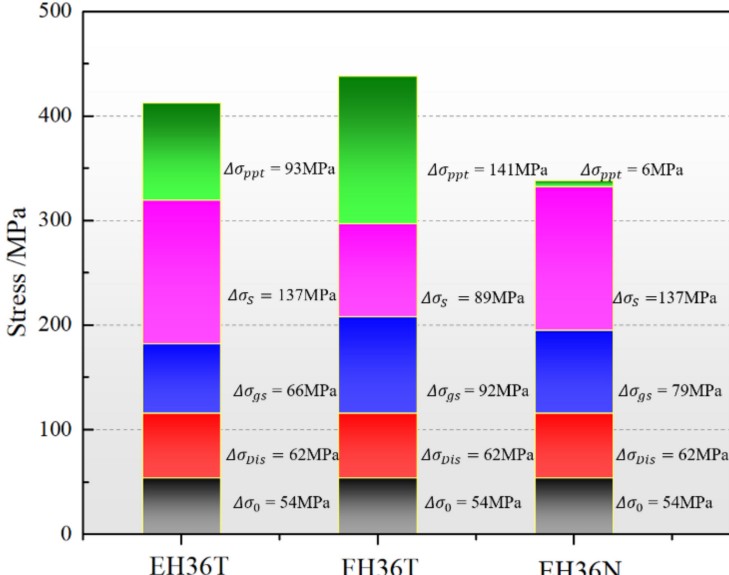

**Figure 9.** Contribution to the yield stress of the different strengthening mechanisms.

### 3.3.3. Improvement in Ductility

Generally, elongation of metallic materials decreases with an increase in strength. However, FH36T had the higher yield stress and elongation compared to that of EH36T. Elongation obtained by the uniaxial tensile test includes uniform elongation (UEL) and local elongation (LE). UEL accounts for the greater part of total elongation [32]. In order to increase UEL, two factors play a significant role. One is the work-hardening rate (n value), whose higher value is in favor of improving the uniformity of the distribution of the stress in the tensile samples. The other one is the ability to sacrifice the volume of the material in the width and thickness direction to make up for the loss brought by the increase of sample length [33,34].

The variation of the n values with an engineering strain of the three cases is shown in Figure 10. It can be seen that EH36T had the highest work hardening rate. Logically, this would be manifested as the highest ductility case; however, as noticed, the elongation of EH36T was the lowest. Thus, there must be another mechanism that explains this behavior.

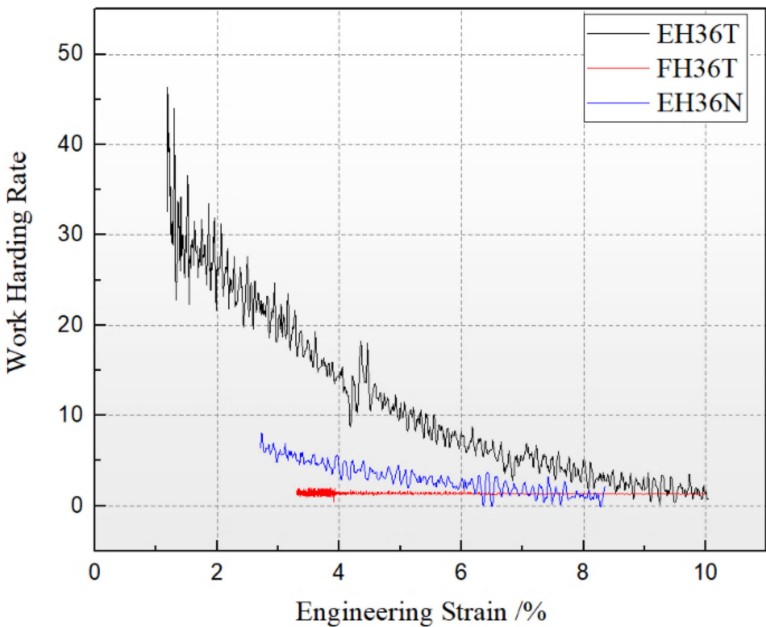

**Figure 10.** Work hardening rates of the offshore structural steel.

The Schmidt factor (SF) can be used to evaluate the ability of material moving in width and thickness directions [34]. A higher of SF value means an easier activation of the crystallographic plane, usually leading to a better ductility and formability for metals. The SFs were calculated as follows:

$$m = \cos \alpha \times \cos \lambda \tag{7}$$

where m is value of the Schmidt factor.

This paper investigated the SFs of close-packed <110> crystallographic planes along the rolled direction (RD), transverse direction (TD), and normal direction (ND) based on their typical deformation textures, which are shown in Table 5. From the results, it can be seen that the SFs of EH36N and FH36T were higher in the TD, which makes it feasible for the tensile samples to compensate for the loss of increase of their length via a decrease in width. In addition, the intensity of the ODFs of EH36N and FH36T were much higher than that of EH36N; therefore, EH36N and FH36T had more grains with favorable orientations of high SFs along the TD.

**Table 5.** SFs of the typical deformation textures of the rolled offshore structural steels.

| Sample | Deformation Texture | SFs | | |
|---|---|---|---|---|
| | | RD | ND | TD |
| EH36T | {112}<110> {554}<225> | 0.41, 0.43 | 0.41, 0.33 | 0.41, 0.41 |
| FH36T | {221}<321> {776}<212> | 0.47, 0.41 | 0.41, 0.32 | 0.48, 0.42 |
| EH36N | {111}<321>,{001}<322> {221}<211> | 0.47, 0.36, 0.41 | 0.27, 0.41, 0.41 | 0.46, 0.41, 0.49 |

*3.4. Fatigue Crack Propagation Analysis*

The fatigue failure process of materials is mainly composed of two key links: crack initiation and propagation. For large-scale engineering components, microcracks inside the specimen are difficult to avoid. Therefore, the study of fatigue crack propagation behavior is very important in engineering applications.

The generally accepted fatigue crack growth theory is the empirical relationship between the crack growth rate $da/dN$ and the stress field intensity factor amplitude proposed by Paris and shown in Formula (8) [19].

$$da/dN = C(\Delta K)^m \tag{8}$$

the values of the constants, C and m, were calculated from the stage II regime of the curves and summarized in Table 6.

**Table 6.** Paris formulas of the offshore structural steels.

| Samples | Paris Formula |
|---|---|
| EH36T | $da/dN = 1.97 \times 10^{-9} \times (\Delta K)^{3.33}$ |
| FH36T | $da/dN = 3.49 \times 10^{-9} \times (\Delta K)^{3.12}$ |
| EH36N | $da/dN = 1.96 \times 10^{-9} \times (\Delta K)^{3.35}$ |

The curves of the fatigue crack growth rate (FCGR measured by $da/dN$) vs. the stress intensity factor amplitude $\Delta K$ are plotted in Figure 11. It can be observed that at the I zone the FCGR of the three offshore structural steels was almost the same. When the $\Delta K$ was higher than 20 MPa/m$^{1/2}$, the FCGR of FH36T was apparently lower than that of EH36T and EH36N, while the FCGR of EH36T was slightly higher than that of EH36N in the $\Delta K$ range from 22.5 to 35 MPa/m$^{1/2}$.

The fracture surfaces of the crack initiation zone (I zone), stable propagation stage (II zone), and unstable propagation stage (III zone) are shown in Figure 12. All the cases show a typical transgranular cleavage surface characterized by a river pattern and a cleavage step. Under higher magnification, fatigue striations can be obviously observed. The "plastic passivation models" proposed by Laird and Krause can be used to explain that the fatigue striations for each propagation of the crack have to pass through the process of "opening, passivation, expansion, sharpening" [35]. A fatigue striation represents a stress cycle process, and the spacing of the fatigue striations represents the distance of the crack propagation of the stress cycle, which can reflect the fatigue crack propagation rate [36].

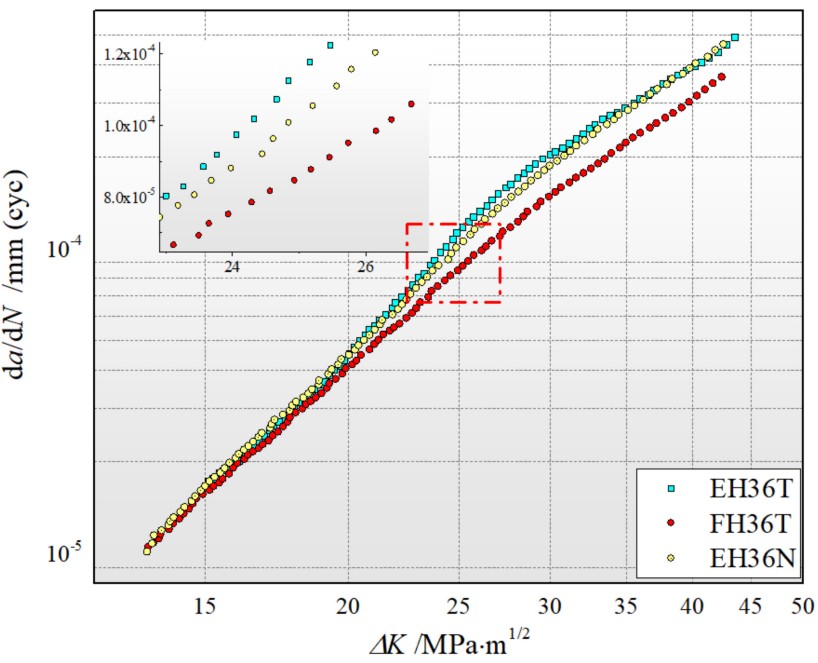

**Figure 11.** Fatigue crack growth behavior of offshore structural steel plates.

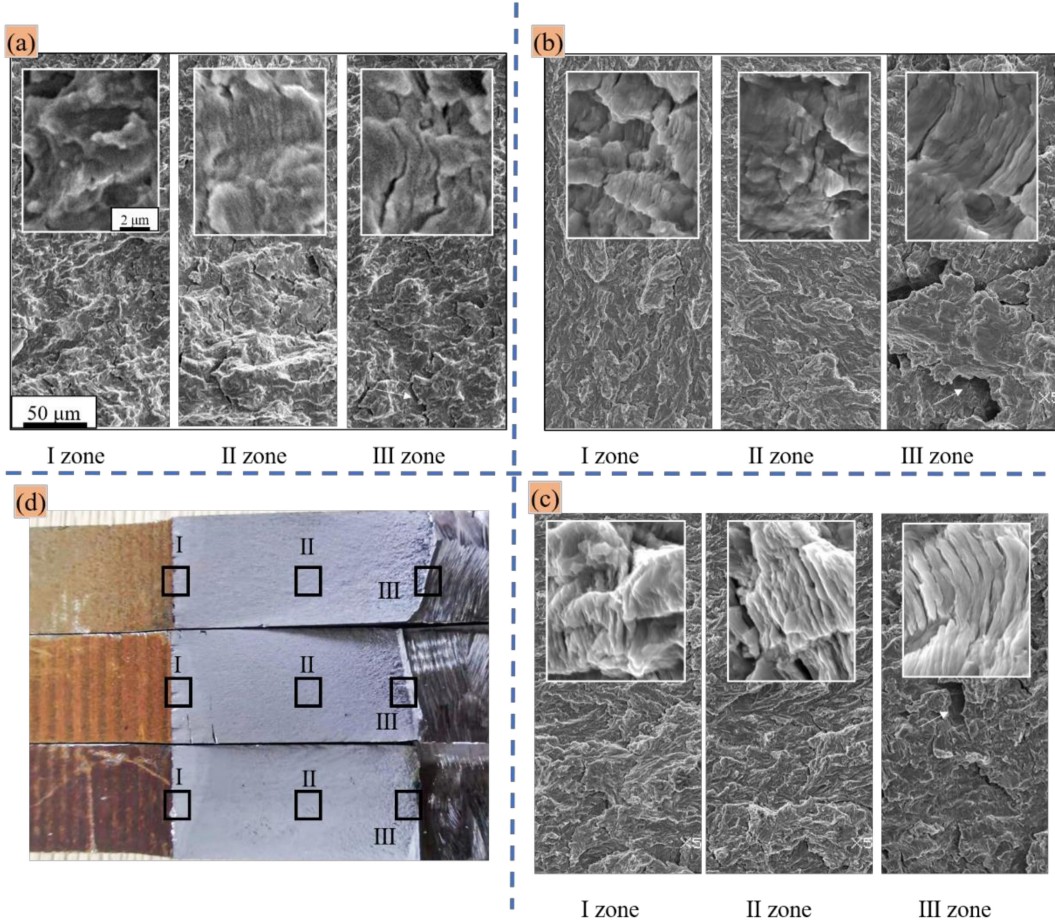

**Figure 12.** Scanning electron micrographs of the fatigue fracture surfaces: (**a**) EH36T, (**b**) FH36T, (**c**) EH36N and (**d**) image of fatigue fractures.

Corresponding to the higher Paris zone, the mechanism of pure fatigue crack propagation is controlled by the double slip mechanism, which is mainly caused by the passivation of the crack tip. The transition from the mechanism of transgranular fracture along the crystallographic plane to the fatigue band fracture mechanism controlled by the double slip mechanism usually occurs in the low Paris region, which is a common phenomenon in superalloys [37]. It has been reported that the local refined grain structure will decline the FCGR by hindering the growth of fatigue crack [38]; however, Birkbeck et al. found no significant change in fatigue crack propagation rate from the change of grain size for low carbon steel [39], and Yakushiji reported that the deformation texture had little influence in FCGR [40]. Thus, the slight deceleration of growth of FCGR for the EH36N specimen compared to that of EH36T can be explained. In terms of FH36T, although its grain size was similar to that of EH36T, at a more microscopic level, the grain boundaries of FH36T showed an obvious difference, as its grain boundaries presented a serrated shape as shown in Figure 5. This kind of grain boundary has been reported to induce local deflection of the growth path of fatigue cracks, thereby decreasing the effective stress intensity factor and further decreasing the FCGR [41,42]. It is worth noting that the deflection can be microscopic, and the variation of the crack path will roughen the fracture surface, which can be observed in Figure 12b, where the fracture of the FH36T sample had the highest roughness. Moreover, the crack closure effect can be significantly enhanced due to the improved residual stress per cycle from the coarse fracture surface, which further increases the fatigue crack resistance.

## 4. Conclusions

This paper developed new kind of offshore structural steel rolled plate product, FH36T (prepared by TMCP), based on micro-adding Cu and Cr elements, and compared the microstructure and mechanical performance of EH36T (EH36 rolled plates prepared by TMCP) and EH36N (manufactured via rough rolling and normalization). The above investigation led to the following conclusions:

(1) The addition of Cu-Cr elements significantly improved the ductility, which increased by 52.2% compared to EH36T, and both yield strength and ultimate strength were enhanced slightly with the values of 438 MPa and 553 MPa, respectively. The serrated shape of the grain boundaries and the precipitation strengthening were the main contributions to the yield stress for the FH36T plate. The normalizing process led to a decrease in yield stress for EH36T, but substantially reduced the yield ratio and increased the ultimate elongation (33%), owing to the sharp decrease of the contribution of precipitation strengthening.

(2) The considerable improvement in elongations of FH36T and EH36N were analyzed by the deformation textures. The results showed that SFs of the deformation texture of FH36T and EH36N in TD and ND were much higher than that of EH36T. This enhances the ability to sacrifice the volume of material in the width and thickness direction to make up for the loss brought by the increase of sample length, thus leading to higher ductility.

(3) FH36T plates showed a better resistance to fatigue crack propagation when $\Delta K$ was higher than $20 \, \mathrm{MPa/m^{1/2}}$ in comparison with the other two offshore steels. This could be attributed to the formation of jagged shape grain boundaries, which decreases the effective stress intensity factor during the fatigue crack propagation process.

**Author Contributions:** Formal analysis, K.H.; funding acquisition, P.Z.; investigation, X.P.; methodology, K.H.; resources, P.Z. and G.L.; visualization, K.H.; writing—original draft, K.H.; writing—review and editing, L.Y. All authors have read and agreed to the published version of the manuscript.

**Funding:** This study was supported in part by a grant from University of Science and Technology Liaoning-State Key Laboratory of Metal Material for Marine Equipment and Application Joint Fund (Grant No. HGSKL-USTLN (2020) 07).

**Data Availability Statement:** Data are available in a publicly accessible repository.

**Acknowledgments:** The authors are grateful for characterization measurements performed by Endian Fan at University of Science and Technology Beijing.

**Conflicts of Interest:** The authors declare no conflict of interest.

## Nomenclature

| | |
|---|---|
| TMCP | thermo-mechanical control process |
| FH36 | Cu–Cr microalloyed offshore structural steel based on EH36 |
| FH36T | TMCP treated FH36 |
| EH36N | normalizing treated EH36 |
| EH36T | TMCP treated EH36 |
| TD | transverse direction |
| ND | normal direction |
| SFs | Schmidt factors |
| YS | yield strength |
| UTS | ultimate tensile strength |
| E | elongation to failure |
| YR | yield ratio |
| $a$ | length of fatigue crack |
| $\Delta K$ | intensity factor |
| $da/dN$ | fatigue crack growth rate |
| $\sigma_y$ | yield strength contributed by different strengthening mechanisms |
| $\Delta\sigma_S$ | yield strength contributed by solid solute strengthening |
| $\Delta\sigma_{gs}$ | yield strength contributed by grain boundary strengthening |
| $\Delta\sigma_{Dis}$ | yield strength contributed by dislocation strengthening |
| $\Delta\sigma_{ppt}$ | yield strength contributed by precipitation strengthening |
| $d$ | average grain size |
| [X] | the weight percent of alloying element X |
| $\rho$ | dislocation density |
| M | Taylor factor |
| G | shear modulus |
| b | Burgers vector |
| s | density of "ridges" |

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
