# Peer review of "Simultaneous Improvement in Mechanical Properties and Fatigue Crack Propagation Resistance of Low Carbon Offshore Structural Steel EH36 by Cu–Cr Microalloying"

_metals, doi:10.3390/met11111880_

Round 1
Reviewer 1 Report
The paper is well written and the results are clearly presented. I have only few concerns which I list here:
1) In the abstract there are a lot of acronysm such as FH36, Cu-Cr, TD, ND, EBSD.... It's better to define each of that ones before using them.
2) the words in fig.3 are too small.
3) Remove '' in the 'conflits of interest' section
Author Response
Comments and Suggestions for Authors
The paper is well written and the results are clearly presented. I have only few concerns which I list here:
1) In the abstract there are a lot of acronysm such as FH36, Cu-Cr, TD, ND, EBSD.... It's better to define each of that ones before using them.
Response:Thank you for your careful review. Cu-Cr are elements and other abbreviations are all refined.
2) the words in fig.3 are too small.
Response:Fig. 3 has been reconstructed.
Figure 3. Specification of fatigue crack propagation test.
3) Remove '' in the 'conflits of interest' section
Response:the “ has been removed

Reviewer 2 Report
More specific comments are described in the comments to the authors as follows.
From an initial glance, the quality of some figures in the manuscript is low (consider to use EMF, EPS, or TIF figure format), the figure background, for example, Fig. 6. Check the data results of the simulation. Also, present properly figures from 16 to 46. Revise the scales and markers used to develop the simulations. Figures should be presented in the same format. Please, revise the author guidelines.
Please check similarity coincidences with https://doi.org/10.1016/j.msea.2021.141426
The article is well written, easy to read; however, some minor grammatical errors were observed without compromising the text as a whole.
Abstract:
Please describe what the problem is and then explain. The abstract should answer the questions: What problem did you study and why is it important? What methods did you use? What were your main results? And what did you conclude from your results? Please make your abstract with specific and quantitative results to further enrich the content of the article. Please indicate each of these questions in the abstract separately when you are replying to this comment.
+ Revise the keywords of the paper.
Introduction:
Last paragraph of Introduction should explicitly explain the gap of the studies reviewed above and the contribution of this manuscript. What is the gap? what is the contribution? These geometries have not been analyzed?
The authors should mention clearly the problems that this study aims to address. What is the scientific challenging in this work in which the authors would like to address (i.e. what are the major differences between this work and previous works)?
Although a comprehensive literature review in the introduction, they were not well-organized. The objectives of the manuscript are also unclear. The authors need to clarify the novelties of this work. The authors should revise the Introduction substantially.
The introduction section should be concise. Many unnecessary details were given while reviewing other people studies, which may not be very interesting to the review.
Why the authors do not talk about of other thermal or thermochemical treatments. I recommend the follow reference: https://doi.org/10.1007/s12666-020-02174-6 - https://doi.org/10.1007/s40735-021-00511-w
Methodology and methods:
+ Please, provide a reference for Fig. 1.
+ Figure 2 mentioned in line 119 is not reported in the paper.
+ Explain in detail how the fatigue crack was calculated. What others kind of models exists to calculate this mechanical property of the materials?.
+ How many repetitions were performed for each experimental condition?
+ Please, provide more information about the experimental conditions used for each test. By i.e. SEM/EDS- EBSD. Intensities, software’s, etc.
Results and discussions:
+ Explain more details about the selection of the EH36 and FH36.
+ The authors should explain their results in detail and deep. There are many possible reasons for the improvement of the mechanical properties, but the authors did not explain the results clearly.
+ Provide more references to compare, discuss and validate the results obtained in a deep way.
+ From the EDS in Figure 4, the authors can be presented only the highest signal on the spectrum, to evidence the contribution of the representative elements of the material.
+ From figure 8, what is the scale of the micrographs?
+ For a better understanding, I think that a list of symbols can be proposed by the authors.
+ According to other mechanical properties of the materials tested, what about the fracture toughness values and how contribute to the industrial application of the material? Please comment on it.
+ Comment about the performance of the fracture toughness of ferrous and ceramic materials related to this study.
+ From Fig. 8, are micrographs or fractographs. Check
Conclusions: The Conclusions could be rewritten according to the activities developed in a depth way. You can use percentages, main findings, etc.
Author Response
More specific comments are described in the comments to the authors as follows.
From an initial glance, the quality of some figures in the manuscript is low (consider to use EMF, EPS, or TIF figure format), the figure background, for example, Fig. 6. Check the data results of the simulation. Also, present properly figures from 16 to 46. Revise the scales and markers used to develop the simulations. Figures should be presented in the same format. Please, revise the author guidelines.
Please check similarity coincidences with https://doi.org/10.1016/j.msea.2021.141426
Response: Thank you for your careful review. Fig. 6 has been rebuilt as clear as possible.
It is necessary to note that no simulation is conducted in this paper. Fig 6 is pole figures calculated from the EBSD data, which is the original experimental data. The quality of the images can not be improved because it is caused by the analysis software. Many papers used channel 5, but this work used OIM, we think it is the reason that the quality of Fig 6 seems not as good as the other SCI papers. Now, Fig. 6 is clear enough to distinguish the deformation texture, and I also want to further improve the image quality, however it is not a technically solvable problem.
All the number in the figures have been modified according to the requirement.
Figure 6. Deformation textures of the studied samples represented by pole figures and ODF section.
(a) EH36T; (b) FH36T; (c) EH36N.
Response: The paper https://doi.org/10.1016/j.msea.2021.141426 is about Ti welding, it has no similarity exist between the two papers.
The article is well written, easy to read; however, some minor grammatical errors were observed without compromising the text as a whole.
Abstract:
Please describe what the problem is and then explain. The abstract should answer the questions: What problem did you study and why is it important? What methods did you use? What were your main results? And what did you conclude from your results? Please make your abstract with specific and quantitative results to further enrich the content of the article. Please indicate each of these questions in the abstract separately when you are replying to this comment.
Response: Thank you for valuable advice. The modified abstract is as-follows:
Improving the mechanical performance of low-carbon offshore steel is of great significance in the shipbuilding applications. In this paper, a new Cu-Cr micro-alloyed offshore structural steel (FH36) is developed based on EH36 [What problem did you study and why is it important?]. The microstructure, mechanical properties and fatigue crack propagation properties of rolled plates of FH36、EH36 and normalizing rolled EH36 plates (EH36N) manufactured by thermo-mechanical control process (TMCP) were analyzed and compared [What methods did you use?]. FH36T shows obvious advantage in elongation with the value of 29%, 52.2% higher than EH36T plates. The normalizing process although led to a relatively lower yield stress (338 MPa), but substantially increased the elongation (33%) and lessened the yield ratio from 0.77 to 0.67. [ What were your main results?] Electron Back-Scattered Diffraction (EBSD) analysis showed SFs of the deformation texture of FH36T and EH36N along transverse direction (TD) and normal direction (ND) are much higher than those of EH36T plate, which enhances the lateral movement ability in width and thickness direction, enhancing the ductility [what did you conclude from your results?]. Moreover, FH36 plates showed a better fatigue crack propagation resistance than rolled EH36 plates [main results 2]. The formation of the jagged shape grain boundaries is believed to induce a decrease of effective stress intensity factor during fatigue crack propagation process[what did you conclude from your results? fatigue crack propagation rate is hard to describe quantitatively in abstract].
+ Revise the keywords of the paper.
Response: Thank you for the careful review, this mistake has been corrected.
Introduction:
Last paragraph of Introduction should explicitly explain the gap of the studies reviewed above and the contribution of this manuscript. What is the gap? what is the contribution? These geometries have not been analyzed?
Response: Thank you for valuable advice. We have explained the gap of studies in introduction, and in the last paragraph, we supplemented the studying content and the contribution. The Last paragraph has been modified as follows:
“Based on the mentioned above, this paper aims at improving mechanical properties via adding welding friendly elements (Cu and Cr), and attempts to further suppress the carbon content to lower than 0.06 wt% to further improve the weldability. The microstructure, mechanical properties, and fatigue crack propagation properties of the newly developed offshore steel are analyzed in comparison with EH36 manufactured by thermo-mechanical control process (TMCP) and normalizing. The result can provide theoretical and data support for future alloy design and processing technique development.”
The authors should mention clearly the problems that this study aims to address. What is the scientific challenging in this work in which the authors would like to address (i.e. what are the major differences between this work and previous works)?
Although a comprehensive literature review in the introduction, they were not well-organized. The objectives of the manuscript are also unclear. The authors need to clarify the novelties of this work. The authors should revise the Introduction substantially.
Response: Thanks for valuable advice, we have modified the introduction section. It is needful to note the paper’s objectives and novelties is to improve mechanical properties via adding elements (Cu and Cr) that are not harmful to weldability, and attempts to further suppress the carbon content to lower than 0.06 wt% to further improve the weldability. That have been clearly addressed in last paragraph. While literature about improving mechanical properties of EH36 by microalloying are relatively rare. Thus, in introduction section we listed some typical alloying elements, pointing their functions and limitation in EH36 plates. On the other hand, as TMCP is an important processing process for preparing offshore structural steel, we gave a brief introduction of this technique.
The introduction section should be concise. Many unnecessary details were given while reviewing other people studies, which may not be very interesting to the review.
Response: yes, we have deleted Irrelevant references.
Why the authors do not talk about of other thermal or thermochemical treatments. I recommend the follow reference: https://doi.org/10.1007/s12666-020-02174-6 - https://doi.org/10.1007/s40735-021-00511-w
Response: We will not only add the two reference, some papers about boriding technique on stainless steel is also available such as “https://doi.org/10.1016/j.triboint.2021.106885”, “https://doi.org/10.1016/j.triboint.2021.107161” and “https://doi.org/10.1016/j.matlet.2020.128842” as follows:
“The manufacturing process plays an important role in realizing the impact of alloy design on steel such as heat-treatment [6, 7] and chemical treatment process [8-10].”
Methodology and methods:
+ Please, provide a reference for Fig. 1.
Response: We have modified Fig. 1, and Fig. 1 is original image based the process based on the material preparation technology, reference is not necessary.
+ Figure 2 mentioned in line 119 is not reported in the paper.
Reponse:Thank you for the careful review, Figure 2 has been added.
+ Explain in detail how the fatigue crack was calculated. What others kind of models exists to calculate this mechanical property of the materials?.
Reponse: We added some detail operation in obtaining as follows:
“To obtain the Paris formula of the fatigue crack growth rate of offshore steels at room temperature, fatigue crack growth rate test of stress-intensity increasing test was carried out based on GB/T 21143-2014 at room temperature. The size of a relevant sample is shown in Figure 3, with the crack propagation direction of the sample along rolled direction. The room temperature fatigue crack growth rate test was carried out using the Instron 8801 hydraulic fatigue testing systems, and the maximum load of the equipment is ±100 kN. During the experiment, a COD gauge was installed on the sample, and the crack length was measured by the flexibility method. The gauge length of the COD was 5 mm and the maximum opening displacement was 2 mm. Fatigue cracks are firstly prefabricated in room temperature. The loading method is axial loading, with the stress ratio of 0.1, maximum force 16 kN and the test frequency of 20 Hz. The smaller cracks in the plastic zone are obtained by step-by-step load reduction. The constant load control is adopted, and the load reduction amplitude of adjacent levels does not exceed 20%. Crack propagation length is about 2 mm. Finally, load is kept constant until the sample breaks to obtain the da/dN and △K. The data points that do not satisfy W-a>4/(Kfmax/Rp0.2)2 are removed to obtain the double logarithmic curve of da/dN~△K, and the Paris formula [25] is obtained through double logarithmic linear fitting by software origin 2016.”
About other kind of models:
In 1963, P.C.Paris give the famous relational expression of the crack propagation law, Paris formula, , which is used in this paper.
To extend the approach, Foreman suggested an equation that brings the crack-propagation rate da/dN to infinite when approaching the fracture toughness Kc and accounts for the mean stress by including the stress ratio R:
McEvily not only took into account the mean stress by the maximum stress-intensity factor Kmax and the fracture toughness Kc, but also the threshold range of the stress intensity factor ΔKth:
Then based on Elber’s observation of crack-closure effects, several plasticity-, roughness-, transformation- or oxide-induced effects may lead to a volume increase in the wake of a propagating crack, giving rise to premature contact of the crack faces when unloading a sample, and hence reducing the crack driving force to an effective range of the stress intensity factor ΔKeff = Kmax – Kop. Below the crack-opening stress intensity factor Kop the crack is closed and the respective residual range of the stress intensity factor ΔKres = ΔK – ΔKeff does not contribute to crack advance:
This article explains the change mechanism of fatigue crack growth rate from the perspective of material microstructure, instead of systematically studying the damage mechanism of materials. The Paris formula is enough to evaluate the fatigue crack growth rate of the material. Such as articles https://doi.org/10.1016/j.engfailanal.2020.105034 .
+ How many repetitions were performed for each experimental condition?
Response:3 times for each tensile and fatigue test. We also address this point in method section.
+ Please, provide more information about the experimental conditions used for each test. By i.e. SEM/EDS- EBSD. Intensities, software’s, etc.
Response:Thank you for the careful review, all the information have been added in the “2. Materials and Methods” section marked by yellow background as follows:
“SEM and EBSD method was carried out by the FEI Quanta Tescan MIRA 3 LMH scanning electron microscope equipped with Oxford EBSD probe at room temperature. The EBSD data is analyzed by OIM software.”
Results and discussions:
+ Explain more details about the selection of the EH36 and FH36.
Response:Selection of the EH36 and FH36 has been well explained in introduction section. This paper carried out a comparative experiment between the two materials.
+ The authors should explain their results in detail and deep. There are many possible reasons for the improvement of the mechanical properties, but the authors did not explain the results clearly.
Response:Yes, there are many possible reasons for the improvement of the mechanical properties. While, the improvement of strength may be different from other mechanical properties ie. wear resistance. Different characterization experiments can obtain different perspectives on material properties. In terms of the strength, we use the structure-property models to explain the strengthening mechanism, which includes most strengthening mechanisms, it is the common method that has been often used in some top journal like “acta materialia”.
In terms of ductility, we program to calculate the effect of slip system, which is redevelopment based on EBSD. It is also a very new method.
In terms of fatigue crack, we have explained the results as possible as we can.
Maybe the result of experiment is not so outstanding, but the relevant analysis is innovative enough.
If you still think it need to add any explanation of improvement of the mechanical properties, we will be very happy to adopt your detail suggestions.
+ Provide more references to compare, discuss and validate the results obtained in a deep way.
Response: The relevant study is rare, however, we still found 1 reference in discussion section, which is [16].
+ From the EDS in Figure 4, the authors can be presented only the highest signal on the spectrum, to evidence the contribution of the representative elements of the material.
Response: Thank you for the valuable advice, the original data of spectrum is in form of picture, which is hard to modify. We change the image into a list, making the data easier and clearer to read. Shown as Table 2
+ From figure 8, what is the scale of the micrographs?
Response: Thank you for the careful review, the scale of Fig. 8 has been added.
+ For a better understanding, I think that a list of symbols can be proposed by the authors.
Response: Yes, this is a very valuable suggestion, and we have built a list of symbols.
Nomenclature and abbreviation
TMCP |
thermo-mechanical control process |
FH36 |
Cu-Cr micro-alloyed offshore structural steel based on EH36 |
FH36T |
TMCP treated FH36 |
EH36N |
normalizing treated EH36 |
EH36T |
TMCP treated EH36 |
TD |
transverse direction |
ND |
normal direction |
SFs |
Schmidt Factors |
YS |
yield strength |
UTS |
ultimate tensile strength |
E |
elongation to failure |
YR |
yield ratio |
a |
length of fatigue crack |
△K |
intensity factor |
da/dN |
fatigue crack growth rate |
σy |
yield strength contributed by different strengthening mechanisms |
ΔσS |
yield strength contributed by solid solute strengthening |
Δσgs |
yield strength contributed by grain boundary strengthening |
ΔσDis |
yield strength contributed by dislocation strengthening |
Δσppt |
yield strength contributed by precipitation strengthening |
d |
average grain size |
[X] |
the weight percent of alloying element X |
r |
dislocation density |
M |
Taylor Factor |
shear modulus |
+ According to other mechanical properties of the materials tested, what about the fracture toughness values and how contribute to the industrial application of the material? Please comment on it.
+ Comment about the performance of the fracture toughness of ferrous and ceramic materials related to this study.
Response: Fracture toughness is very important for materials. However, this paper mainly focuses on the effect of microstructural evolution on tensile strength and fatigue crack propagation resistance of offshore structure steel. If fracture toughness or other mechanical properties is added, new characterization should be also carried out. We are also very interested in the effect of microstructure on fracture toughness or other mechanical properties, but we think it is more suitable to build a new article on the fracture toughness. Moreover, editors only give us 3 days to revise this paper, it is not possible to complete the extra experiments.
ceramic materials have many advantages against metal such as the superior insulativity and abrasive resistance. I am also very interested in this field, but in this paper, appearance of ceramic materials seems not very suitable.
We will try our best to improve the quality of this article, and revise this paper according to the reviewer’s requirement including adding useful references and recompose the paper’s structure etc. However, we have to make the necessary compromises for which is beyond our ability. I sincerely hope you can understand our difficult situation.
+ From Fig. 8, are micrographs or fractographs. Check
Response: Fig. 8 shows fractographs of tensile samples.
Conclusions: The Conclusions could be rewritten according to the activities developed in a depth way. You can use percentages, main findings, etc.
Response: Thanks for the valuable advice, as Conclusions (2) and (3) are qualitative analysis conclusions, thus we only rewritten the (1) in conclusion as follows:
(1) The addition of Cu-Cr elements significantly improves the ductility, which increases by 50.2% compared to EH36T, and both yield strength and ultimate strength are enhanced slightly with the values of 438 MPa and 553 MPa, respectively. The serrated shape of grain boundaries and the precipitation strengthening is the main contributions to the yield stress for FH36T plate. The normalizing process led to a decrease in yield stress for EH36T, but substantially reduced the yield ratio and increased the ultimate elongation (33%), which is owing to the sharply decrease of contribution of precipitation strengthening.

This manuscript is a resubmission of an earlier submission. The following is a list of the peer review reports and author responses from that submission.